# Effect of the COVID-19 pandemic on health facility delivery in Ethiopia; results from PMA Ethiopia's longitudinal panel

**Linnea A. Zimmerman**[1]*, **Selamawit Desta**[1], **Celia Karp**[1], **Mahari Yihdego**[2], **Assefa Seme**[3], **Solomon Shiferaw**[3], **Saifuddin Ahmed**[1]

**1** Department of Population Family and Reproductive Health, Johns Hopkins Bloomberg School of Public Health, Baltimore, Maryland, United States of America, **2** PMA-Ethiopia, Addis Ababa University, Addis Ababa, Ethiopia, **3** School of Public Health, Addis Ababa University, Addis Ababa, Ethiopia

* linnea.zimmerman@jhu.edu

**Data Availability Statement:** All data used in these analyses are publicly available and can be accessed at https://www.pmadata.org/data/request-access-

## Abstract

### Objectives

To examine the effect of COVID-19 on health facility delivery in Ethiopia.

### Design

We used longitudinal data with a pre-post exposure to the pandemic to assess change in facility delivery patterns nationally and by urban and rural strata. We conducted design-based multivariable multinomial logistic regression comparing home, lower-level facility, and hospital delivery with date of birth as a spline term, with a knot indicating births before and on/after April 8, 2020.

### Setting

Six regions in Ethiopia, covering 91% of the population.

### Participants

Women were eligible to participate if they were currently pregnant or less than six weeks postpartum and were recontacted at six weeks, six months, and one year postpartum. 2,889 women were eligible and 2,855 enrolled. Data used in this paper come from the six-week interview, with a follow-up rate of 88.9% (2,537 women).

### Results

In urban areas, women who delivered during the COVID-19 pandemic had a 77% reduced relative risk of delivering in a hospital relative to women who delivered prior to the pandemic (aRRR: 0.23, 95% CI: 0.07–0.71). There were no significant differences between the pre- and COVID-19 periods within rural strata where the majority of women deliver at home (55.6%). Overall, the effect was non-significant at a national level. Among women who delivered during the COVID-19 pandemic, 20.0% of urban women said COVID-19 affected where they delivered relative to 8.7% of rural women (p-value = 0.01).

datasets. The doi for the dataset is https://doi.org/0.34976/8r5s-dx31.

**Funding:** This work was supported, in whole, by the Bill and Melinda Gates Foundation [INV 009466]. LAZ was the recipient of the grant. The funders had no role in study design, data collection and analysis, decision to publish, or preparation of the manuscript.

**Competing interests:** The authors have declared that no competing interests exist.

## Conclusion

We found that delivery patterns in urban areas changed during the early months of the COVID-19 pandemic, but there was no evidence of large-scale declines of hospital delivery at the national level. Concerns about COVID-19 transmission in health facilities and ensuring lower-level facilities are equipped to address obstetric emergencies are critical to address. COVID-19 will likely slow progress towards increasing rates of institutional delivery in urban areas in Ethiopia.

## Introduction

Improving the proportion of births attended by skilled health personnel, predominately through increasing the proportion of births delivered in a health facility that offers obstetric care, is a key intervention strategy to reduce high maternal mortality in low- and middle-income countries [1, 2] and is recognized as an indicator for achieving the Sustainable Development Goal-3 (SDG indicator 3.1.2). Model based estimates of COVID-19's impact on MNH have been ominous, due to anticipated declines in coverage of services from overburdened health systems, restrictions on movement, and reductions in care-seeking [3–5]. Roberton and colleagues estimated that a reduction of 10–19% in coverage of key maternal and newborn health interventions, combined with increases of 10% in infant wasting for six months, could result in 253,000 additional child deaths and 56,700 maternal deaths across 118 countries [3]. Similarly, estimates from the Guttmacher Institute suggest a 10% reduction in coverage of essential pregnancy and newborn-related care would result in 28,000 additional maternal and 168,000 additional newborn deaths, globally [5].

While early studies based on prediction models provided valuable insight into the potential impact of the pandemic, the models are mostly based on a set of assumptions in the absence of empirical data. This is especially challenging in low- and middle-income countries where population-level data on the effects of COVID-19 on MNH care-seeking and health behaviors are extremely limited, but where evidence from previous pandemics has found serious consequences for maternal and newborn health, including increased maternal and newborn morbidity and mortality and reductions in health seeking behaviors [6–8]. A prospective observational study of pregnant women in Nepal examined data pre- and post-COVID-19 lockdown and identified a reduction in facility births of 52% [9]. In India, stillbirth rates among cohorts of women who delivered during COVID-19 period were found to be significantly higher than among women who delivered prior to COVID-19, primarily due to delays in reaching health care facilities and/or denial of care upon arrival [10]. A facility-based study in Ethiopia identified that fear of COVID-19 and interruption of services due to COVID-19 were associated with lower odds of receiving recommended antenatal care services [11]. While the studies from Nepal and India provide important insight into the potential impact of the pandemic, they are limited in their applicability to Ethiopia or other sub-Saharan African countries given differences in both contexts and the trajectory of the pandemic. The Ethiopian study, the only we could find in sub-Saharan Africa that specifically assessed changes in care-seeking, only included women who successfully accessed antenatal care services—a substantial limitation in understanding the effect of COVID-19 on overall maternal health care seeking.

To date, there are no studies examining the effects of COVID-19 on facility-based deliveries and intrapartum care in Ethiopia. This study uses longitudinal data from a cohort of women

in Ethiopia to assess changes in patterns of maternal health service utilization. Specifically, our research objective is to determine whether patterns of delivery location differed between women who delivered prior to and during the COVID-19 pandemic, at the national level and by urban and rural residence. Additionally, we explore whether women reported that COVID-19 affected where they delivered and sociodemographic correlates of this impact.

## Methods

### Ethics approval statement

PMA Ethiopia received ethical approval from Addis Ababa University, College of Health Sciences (AAU/CHS) (Ref: AAUMF 01–008) and the Johns Hopkins University Bloomberg School of Public Health (JHSPH) Institutional Review Board (FWA00000287).

### Patient consent statement

All participants gave consent to participate and consent procedures approved by the Addis Ababa University, College of Health Sciences (AAU/CHS) (Ref: AAUMF 01–008) and the Johns Hopkins University Bloomberg School of Public Health (JHSPH) Institutional Review Board (FWA00000287).

### Study context

Ethiopia has made remarkable progress in reducing maternal, neonatal, and child mortality in recent years—progress that is now threatened by the global COVID-19 pandemic [12]. Between 2000 to 2016, Ethiopia's under-5 mortality declined from 48 to 28 per 1,000 live births, while the maternal mortality ratio declined from 897 to 323 deaths per 100,000 live births [13]. These improvements in the health of women and newborns have occurred with concomitant increases in key MNH interventions. Facility-based deliveries increased from 5.0% in 2000 to 26.2% in 2016, while delivery by a skilled provider increased from 10.8% in 2011 to 27.7% in 2016 [14, 15].

As of February 10[th], 2021, there were 143,566 confirmed cases of COVID-19 in Ethiopia and 2,158 deaths [16]. A state of emergency was declared on April 8, to help curb the spread of the virus [17]. Under this order, the federal government mandated the closures of schools, closed land borders and airports, and imposed strict restrictions on public and private transportation and movement. Though many restrictions have since been eased, as of February, 2021, social distancing guidelines, mask-wearing mandates, and restrictions against gatherings of more than 100 people remain in place [18].

### Data

Data for this study come from the Performance Monitoring for Action (PMA) Ethiopia survey, conducted in collaboration between Addis Ababa University and Johns Hopkins Bloomberg School of Public Health. PMA Ethiopia is a survey project composed of three separate survey components: an annual cross-sectional survey conducted nationally, a panel survey that follows pregnant women through one year postpartum that is conducted in six regions covering 91% of the population, and annual Service Delivery Point (SDP) surveys. The data for this analysis come from the panel survey of women.

Between October and November 2019, a census was conducted in 217 enumeration areas (EAs) among 36,614 households. All women aged 15–49 were screened (32,792) and, if they reported being currently pregnant or having delivered within the past six weeks, were eligible for the panel study; 2,889 women were identified as eligible and 2,855 enrolled. Each woman

completed a baseline survey at the time of enrollment and will complete follow-up interviews at six-weeks, six-months, and one-year postpartum. Data on labor and delivery used in this paper were reported at the six-week interview, which had a follow-up rate of 88.9% (2,537 women).

Data collection paused in early April due to the COVID-19 pandemic. During the pause in data collection, modifications were made to the questionnaires to include a range of questions about COVID-19 knowledge and risk and the role of COVID-19 in care-seeking behaviors for MNH. When data collection resumed amid COVID-19 in June, with enhanced safety protocols, including social distancing, COVID-19 symptom screening, and mandatory mask requirements, all women with outstanding surveys were interviewed using the updated questionnaires. The COVID-19 pandemic introduced a "natural experiment" within the PMA Ethiopia cohort, providing a unique opportunity to apply a pre-post study design paradigm to examine differences among women who delivered and completed six-week postpartum interviews before versus during COVID-19. Appendix A in S1 Table depicts how women's interviews were affected by the COVID-19 pandemic and the pause in data collection.

## Ethical approval

Oral consent to participate was obtained at the initial household screening, then from each woman for the screening, and finally, prior to enrollment in the panel survey for all eligible women. Oral consent was documented by interviewers within each survey form. All consents were provided as oral consent per guidance from the National Research Ethics Review Guidelines. Based on this guidance and from the IRB on record, written consent is not required in areas of low literacy or when data collection does not include invasive procedures (e.g. biospecimen collection). Additionally, the National Research Ethics Review Guidelines consider women age 15–17 as able to consent for themselves when data collection covers sensitive topics, including sexual and reproductive health, thus no parental consent was required. All procedures, including consent procedures, were approved by both the Addis Ababa University [075/13/SPH] and Johns Hopkins Bloomberg School of Public Health [00009391] Institutional Review Boards. Additional information on the PMA Ethiopia survey, including more detail on the informed consent procedures, can be found at Zimmerman [19].

## Dependent variables

COVID-19 caseloads were, and continue to be, prioritized in hospitals where ICU and ventilators are available; therefore, we sought to examine whether facility-based delivery patterns shifted from hospitals, which are likely to bear the burden of COVID-19, to lower-level health centers during the pandemic. We first assessed women's delivery location—categorized as home delivery, governmental hospital delivery, or delivery in non-hospital health facilities, including in a non-governmental hospital (lower-level facilities)—at the national level (The PMA Ethiopia panel is conducted in 6 regions which together cover an estimated 91% of the population of Ethiopia. We will thus refer to this as the national sample, but acknowledge that 5 regions are not include in the panel and thus, the sample is not completely representative of all women.) and by urban/rural strata. Second, among women who gave birth April 8, 2020 or later, the day the State of Emergency was instated, or later, we assessed the proportion that responded affirmatively to the question, "Did the Coronavirus pandemic affect where you delivered?".

## Independent variables

**Primary independent variable.**   Our primary independent variable of interest was a binary variable, indicating whether the woman delivered prior to or during the COVID-19 pandemic, proxied by the April 8th declaration of the State of Emergency.

## Adjustment variables

We include relevant sociodemographic characteristics in our analysis to identify selection in place of delivery. Specifically, we assessed age (15–19, 20–29, 30+), parity, education, and wealth. We showed estimates at both the national level and by urban and rural strata separately. In national estimates, wealth was treated as a binary variable, indicating whether the woman lived in the poorest 40% of households (coded as 0 - "poor") or the wealthiest 60% of households (coded as 1 –"less poor") nationally. Given that greater than 97% of women in urban areas lived households defined as less poor (Table 1 below), we rescaled wealth for the strata specific estimates. Specifically, in urban areas, wealth is categorized as whether the women lived in the poorest 40% of urban households (coded as 0) or the wealthiest 60% of urban households (coded as 1), while in rural areas, wealth is categorized as whether the women lived in the poorest 40% of rural households (coded as 0) or the wealthiest 60% of rural households (coded as 1). These distinctions allow us to explore the effect of wealth both at the national level and within urban and rural strata separately.

## Analysis

Descriptive statistics assessed the frequency of each outcome in the sample overall and by month. All analyses were performed first among the full sample and then by strata.

**Table 1. Sample characteristics of all women in PMA Ethiopia who completed six-week questionnaire.**

|  | Total | Rural | Urban | p-value |
|---|---|---|---|---|
|  | % | % | % |  |
| Total (unweighted N) | 2,547 | 1,566 | 981 |  |
| Delivery timing |  |  |  |  |
| Pre-COVID-19 | 83.8 | 83.1 | 86.1 |  |
| COVID-19 | 16.2 | 16.9 | 13.9 | 0.07 |
| Delivery location |  |  |  |  |
| Home | 44.7 | 55.6 | 7.7 |  |
| Lower-level facility | 34.7 | 31.9 | 44.1 | ≤0.01 |
| Hospital | 20.6 | 12.5 | 48.3 |  |
| Age |  |  |  |  |
| 15–19 | 10.7 | 11.6 | 7.5 |  |
| 20–29 | 55.0 | 51.1 | 68.1 | ≤0.01 |
| 30+ | 34.4 | 37.3 | 24.4 |  |
| Parity |  |  |  |  |
| 0 | 17.9 | 14.6 | 29.2 |  |
| 1–2 | 38.1 | 34.1 | 51.6 | ≤0.01 |
| 3+ | 44.0 | 51.3 | 19.2 |  |
| Wealth (national) |  |  |  |  |
| Poorest 40% | 40.0 | 51.0 | 2.5 |  |
| Less-poor 60% | 60.0 | 49.0 | 97.5 | ≤0.01 |
| Education |  |  |  |  |
| None | 41.1 | 48.8 | 14.9 |  |
| Primary | 40.0 | 41.1 | 36.3 | ≤0.01 |
| Secondary + | 18.9 | 10.1 | 48.8 |  |
| Residence |  |  |  |  |
| Rural | 77.3 | 100.0 | 0.0 |  |
| Urban | 22.7 | 0.0 | 100.0 |  |

We conducted bivariate and multivariable multinomial logistic regression comparing home delivery (reference category), lower-level facility delivery, and hospital delivery with the date of birth (in months) as a spline with a knot indicating births that occurred before April 8[th], 2020 (pre-COVID) and April 8[th] or after (COVID-19). All analyses were weighted and adjusted for complex survey design and loss-to-follow-up with Taylor linearization method.

Finally, among all women who gave birth during COVID-19, we assessed the characteristics of women who stated that COVID-19 affected their delivery location and provided descriptive statistics of responses describing how COVID-19 affected their place of delivery.

### Patient and public involvement

PMA Ethiopia engaged with federal and regional health bureaus prior to the launch of the survey, including receiving feedback on design of the survey, key indicators, questionnaire design and obtaining letters of support. Community leaders are contacted prior to the launch of the survey. Enumerators for PMA Ethiopia are hired from or near the communities in which the survey is conducted, whenever feasible. Results are disseminated back to federal and regional health bureaus through regular disseminated events.

## Results

### Sample characteristics

Table 1 shows the sample characteristics overall and by residence, with p-values for the design-based F-statistic comparing urban and rural distributions. The percentage of births that occurred during COVID-19 was 16.2% (n = 400) and there were no statistically significant differences by residence. Approximately 45% of all births occurred at home, varying significantly by residence, where 55.6% of rural births occurred at home compared to 7.7% of urban births. Only 12.5% of rural births occurred in a hospital relative to 48.3% of urban births.

Socio-demographic characteristics varied significantly by residence, with women in urban areas more likely to be age 20–29, have fewer previous births, and have attended at least some secondary school. The vast majority of women in urban areas (97.5%) lived in households defined as "less-poor", compared to 49.0% of rural women.

Fig 1 shows the monthly change in delivery location between September 2019 and May 2020, at the national level and for urban and rural strata. Births in May and June are combined in the figure due to the small number of women who gave birth in June. National trends show that hospital births decreased in May/June, while home deliveries and lower-level facility

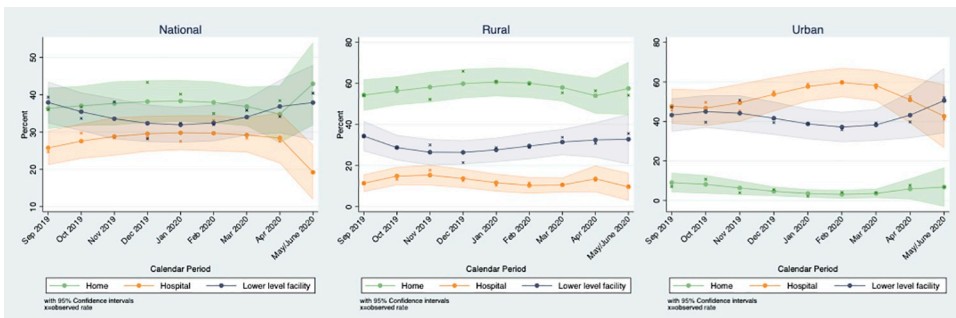

**Fig 1. Percentage of births delivered at home, in a government hospital or in a lower-level health facility by month and residence.**

deliveries increased. The national trends mask significant differences between urban and rural strata, however. In rural areas, trends in delivery location were fairly stable, with home delivery occurring in around 60% of births across all months and hospital delivery occurring in less than 20%. In urban areas, the majority of births took place in hospitals, before declining in April and May while about 40% of births in each month occurred in lower-level facilities, with an increase observed during May and June.

Table 2 shows the crude and adjusted relative risk ratios (aRRRs) for delivering in a lower-level health facility relative to a home delivery and for delivering in a hospital relative to a home delivery in the total sample. There was no difference in the pattern of deliveries between the pre-COVID-19 and COVID-19 periods in the national sample.

The relative risk of delivering in a hospital relative to at home was significantly higher among women age 20–29 and 30 and above relative to women age 15–19 (aRRR: 2.15, 95% CI: 1.24–3.72 and aRRR: 3.51, 95% CI: 1.75–7.04, respectively). Conversely, the relative risk of delivering in either a lower-level facility or a hospital was significantly less for multi-parous women relative to women who were delivering for the first time. Living in a household that was less-poor increased the relative risk of delivering in both a lower-level facility and a hospital (aRRR: 1.66, 95% CI: 1.29–2.12 and aRRR: 2.12, 95% CI: 1.47–3.05, respectively). Having attended secondary school significantly increased the relative risk of delivering in a lower-level facility and attending either primarily or secondary school increased the risk of delivering in a hospital, relative to women who had no education. Urban residence increased the relative risk of delivering in a lower-level health facility by a factor of 5.07 (95% CI: 2.78–9.23) and increased the relative risk of delivering in a hospital by a factor of 10.77 (95% CI: 5.81–19.96).

**Table 2. Adjusted relative risk ratio of delivering in a lower-level health facility or government hospital relative to home delivery among women (national).**

| | Lower Level Facility | | Hospital | |
|---|---|---|---|---|
| | **Crude** | **Adjusted** | **Crude** | **Adjusted** |
| | **RRR** | **RRR** | **RRR** | **RRR** |
| Delivery timing | | | | |
| Pre-COVID-19 | 1.00 (0.94–1.06) | 0.98 (0.91–1.05) | 1.00 (0.94–1.07) | 0.96 (0.88–1.04) |
| COVID-19 | 1.30 (0.83–2.04) | 1.41 (0.85–2.33) | 0.81 (0.45–1.44) | 0.98 (0.51–1.91) |
| Age (ref: 15–19) | | | | |
| 20–29 | 0.9 (0.63–1.3) | 1.24 (0.77–2.02) | 1.47 (0.88–2.45) | 2.15 (1.24–3.72)* |
| 30+ | 0.61 (0.42–0.88)** | 1.74 (0.95–3.19)^ | 0.79 (0.44–1.41) | 3.51 (1.75–7.04)*** |
| Parity (ref: 0) | | | | |
| 1–2 | 0.56 (0.38–0.81)*** | 0.50 (0.32–0.78)** | 0.34 (0.23–0.51)*** | 0.26 (0.16–0.42)*** |
| 3+ | 0.23 (0.17–0.33)*** | 0.26 (0.16–0.44)*** | 0.12 (0.08–0.18)*** | 0.15 (0.08–0.27)*** |
| Wealth (ref: poorest) | | | | |
| Less poor | 3.28 (2.32–4.64)*** | 1.66 (1.29–2.12)*** | 7.49 (4.72–11.89)*** | 2.12 (1.47–3.05)*** |
| Education | | | | |
| Primary | 1.76 (1.33–2.32)*** | 1.25 (0.91–1.71) | 2.81 (2.07–3.81)*** | 1.76 (1.18–2.62)* |
| Secondary + | 6.98 (4.4–11.08)*** | 2.47 (1.42–4.29)** | 18.63 (11.23–30.93)*** | 3.94 (2.2–7.04)*** |
| Residence (ref: rural) | | | | |
| Urban | 10.03 (5.5–18.29)*** | 5.07 (2.78–9.23)*** | 28.12 (14.43–54.8)** | 10.77 (5.81–19.96)*** |

^p<.10
* p<.05
**p<.01
***p<.001.

**Table 3. Adjusted relative risk ratio of delivering in a lower-level health facility or government hospital relative to home delivery among women (urban).**

|  | Lower Level Facility | | Hospital | |
|---|---|---|---|---|
|  | Crude | Adjusted | Crude | Adjusted |
|  | RRR | RRR | RRR | RRR |
| Delivery timing |  |  |  |  |
| Pre-COVID-19 | 1.11 (0.99–1.26)^ | 1.09 (0.95–1.25) | 1.2 (1.06–1.36)* | 1.14 (0.98–1.32)^ |
| COVID-19 | 0.61 (0.17–2.26) | 0.54 (0.17–1.7) | 0.25 (0.07–0.91)* | 0.23 (0.07–0.71)** |
| Age (ref: 15–19) |  |  |  |  |
| 20–29 | 1.34 (0.51–3.56) | 0.93 (0.33–2.57) | 2.23 (0.91–5.48)^ | 1.35 (0.56–3.29) |
| 30+ | 0.45 (0.14–1.48) | 0.45 (0.1–2.13) | 1.00 (0.33–3.04) | 1.12 (0.3–4.14) |
| Parity (ref: 0) |  |  |  |  |
| 1–2 | 0.99 (0.47–2.10) | 1.58 (0.62–4.06) | 0.65 (0.32–1.32) | 1.06 (0.42–2.63) |
| 3+ | 0.33 (0.15–0.72)* | 1.13 (0.33–3.90) | 0.22 (0.10–0.50)*** | 0.71 (0.22–2.23) |
| Wealth (ref: poorest) |  |  |  |  |
| Less poor | 7.28 (3.41–15.52)*** | 5.54 (2.46–12.46)*** | 16.44 (7.46–36.24)*** | 9.95 (4.38–22.59)*** |
| Education |  |  |  |  |
| Primary | 4.53 (1.63–12.61)** | 3.05 (0.92–10.11)^ | 8.24 (2.9–23.42)*** | 5.25 (1.60–17.30)* |
| Secondary + | 6.84 (3.08–15.16)*** | 2.78 (1.22–6.32)* | 20.54 (9.18–45.97)*** | 5.84 (2.59–13.17)*** |

^p<.10

*p<.05

**p<.01

***p<.001.

As shown in Table 3, women in urban areas who delivered during the COVID-19 pandemic had a 77% reduced relative risk of delivering in a hospital relative to women who delivered prior to the pandemic (aRRR: 0.23, 95% CI: 0.07–0.71).

Inequity in maternity care is a continuing concern in Ethiopia. Our analysis suggests women who lived in urban households that were economically less-poor had significantly higher relative risk of delivering in either a lower-level facility (aRRR: 5.54, 95% CI: 2.46–12.46) or a hospital (aRRR: 9.95, 95% CI: 4.38–22.59), compared to the women in the poorest households. Women with more education, particularly women with secondary or higher education, had a higher relative risk of delivering in either a lower-level facility or a hospital.

The crude and adjusted relative risk ratios for delivering in a lower-level health facility relative to a home delivery and for delivering in a hospital relative to a home delivery among rural women is shown in Table 4. On the whole, the pattern in rural areas is similar to the national pattern. There was no significant difference in place of delivery among rural women between the pre-COVID-19 and COVID-19 periods. Older women (age 30 and above) had significantly higher relative risk of delivering in either a lower-level facility or a hospital than women age 15–19, while women age 20–29 had a significantly higher risk of delivering in a hospital. Women with secondary or higher education had significantly higher relative risk of delivering in a lower-level facility or a hospital than at home, while multiparous women were significantly less likely to deliver in either a lower-level facility or a hospital. Living in a less poor rural household increased the relative risk of delivering in a lower-level facility by 1.79 times (95% CI: 1.24–2.57) and the relative risk of delivering in a hospital by 2.18 times (95% CI: 1.31–3.63, respectively).

Among women who delivered during the COVID-19 period, approximately 11% reported that the pandemic affected where they delivered (Table 5). There were no significant differences by socio-demographic characteristics other than residence. The percentage of women

**Table 4. Adjusted relative risk ratio of delivering in a lower-level health facility or government hospital relative to home delivery among women (rural).**

| | Lower Level Facility | | Hospital | |
|---|---|---|---|---|
| | Crude | Adjusted | Crude | Adjusted |
| | RRR | RRR | RRR | RRR |
| Delivery timing | | | | |
| Pre-COVID-19 | 1.01 (0.94–1.08) | 0.98 (0.91–1.06) | 0.96 (0.87–1.06) | 0.90 (0.80–1.01)^ |
| COVID-19 | 1.36 (0.83–2.23) | 1.41 (0.82–2.41) | 1.32 (0.63–2.76) | 1.45 (0.68–3.10) |
| Age (ref: 15–19) | | | | |
| 20–29 | 0.74 (0.48–1.15) | 1.29 (0.74–2.25) | 0.95 (0.51–1.78) | 2.57 (1.35–4.91)** |
| 30+ | 0.63 (0.41–0.97)* | 2.10 (1.06–4.15)* | 0.60 (0.27–1.35) | 3.61 (1.54–8.45)** |
| Parity (ref: 0) | | | | |
| 1–2 | 0.52 (0.34–0.80)** | 0.47 (0.28–0.79)** | 0.28 (0.16–0.49)*** | 0.18 (0.10–0.34)*** |
| 3+ | 0.28 (0.19–0.42)*** | 0.23 (0.13–0.42)*** | 0.17 (0.1–0.3)*** | 0.11 (0.05–0.23)*** |
| Wealth (ref: poorest) | | | | |
| Less poor | 2.09 (1.45–3.01)*** | 1.79 (1.24–2.57)** | 2.61 (1.55–4.41)*** | 2.18 (1.31–3.63)** |
| Education | | | | |
| Primary | 1.45 (1.08–1.96)* | 1.20 (0.86–1.67) | 1.92 (1.34–2.75)*** | 1.42 (0.91–2.21) |
| Secondary + | 4.41 (2.52–7.72)*** | 2.65 (1.39–5.07)** | 5.09 (2.84–9.11)*** | 2.3 (1.09–4.84)* |

^p<.10

*p<.05

**p<.01

***p<.001.

**Table 5. Bivariate association of select characteristics and reporting that place of delivery was affected by COVID-19 among women who delivered April 8th, 2020 and after.**

| | % | F-statistic |
|---|---|---|
| Total | 10.9 | |
| Month of delivery | | |
| April | 8.0 | |
| May/June | 12.5 | 0.16 |
| Residence | | |
| Rural | 8.7 | |
| Urban | 20.0 | 0.01 |
| Age | | |
| 15–19 | 7.3 | |
| 20–29 | 12.6 | |
| 30+ | 9.7 | 0.45 |
| Parity | | |
| 0 | 12.8 | |
| 1–2 | 12.5 | |
| 3+ | 8.4 | 0.48 |
| Wealth (national) | | |
| Lowest 40% | 7.5 | |
| Highest 60% | 13.3 | 0.10 |
| Education | | |
| None | 8.3 | |
| Primary | 10.7 | |
| Secondary + | 17.6 | 0.16 |

who said COVID-19 affected their delivery location was more than twice as high in urban relative to rural areas (20.0% versus 8.67%, p = 0.01).

Among women who said that the pandemic affected where they delivered, 71.3% shared that fear of COVID-19 transmission affected their decision, while 42.4% said they were afraid they would be alone during delivery and 35.9% said that they had no transportation. The complete distribution is given in Appendix A in S1 Table.

## Discussion

Our objective was to determine whether the COVID-19 pandemic affected facility deliveries in Ethiopia and if this varied by urban/rural residence. We observed no significant changes in delivery location at the national level or in rural areas. In contrast, we found significant shifts in delivery patterns in urban areas, with women who delivered during the pandemic being significantly less likely to deliver in a hospital relative to women who delivered prior to the pandemic.

In the months prior to the pandemic, we observed limited changes to facility-based delivery rates and at the national level, did not find evidence to support the assumption of wide-spread declines in facility-based delivery that have been used to inform modeling exercises. Ethiopia has invested heavily in their primary health care system with a focus on MNH and has achieved significant declines in maternal and newborn morbidity and mortality as a result [12]. We found overall levels of facility-based delivery that were significantly higher than the most recent Demographic and Health Survey (DHS) [15]. Though part of these differences can be explained by differences in study design,—the DHS produces estimates that reflect facility-based delivery rates approximately 2.5 years prior to the survey while the PMA Ethiopia survey is designed to produce facility-based delivery rates within the past one year—these numbers also reflect an ongoing trend towards increased facility deliveries. We cannot determine whether coverage of facility-based delivery declined significantly since mid-June 2020, when our data collection was completed, however, it is likely that coverage of interventions was most impacted early on when restrictions were most stringent.

While these trends are encouraging, shifts away from hospital deliveries in urban areas may impose significant health risks for women and their newborns if lower-level facilities are unable to offer the same level of care as hospitals, particularly during obstetric and neonatal emergencies. In 2016, the FMOH found that no health centers or health clinics in the country were equipped to offer all components of comprehensive emergency obstetric and newborn care (CEmONC) and only 5% were able to offer all components of basic emergency obstetric care (BEmONC); the corresponding proportions for hospitals/maternal and child health (MCH) specialty centers were 45% and 14%, respectively [20]. Data from the 2019 PMA Ethiopia SDP survey found that health centers were significantly less likely to have life-saving maternal and reproductive health medicines on-hand on the day of the survey relative to hospitals [21], indicating that these disparities continue. As the COVID-19 pandemic continues, it will be critical to ensure that lower-level health facilities, which are unlikely to address COVID-19 cases, are equipped to address obstetric emergencies and reduce the burden on hospitals.

During the early stages of the pandemic, when restrictions on movement were strictest, only 11% of women in our sample reported that COVID-19 affected where they delivered. Our finding that the proportion of women reporting COVID-19's impact on delivery location differed significantly by urban versus rural residence aligns with our overall findings: in the early months of the pandemic, the COVID-19 had greater impacts on urban areas than rural areas. Though the sample size is limited, the primary reasons that women gave—fear of COVID-19 transmission and being afraid that they would be alone during delivery—are

barriers to utilization of facility-based delivery services that can be effectively addressed by the FMOH through communication and messaging about the safety measures put in place to prevent spread of COVID-19. Limited availability and access through reductions in transportation or facility closures were less frequently cited as reasons for how COVID-19 impacted delivery location, indicating that changes in access and availability of services were not major barriers to care.

Independent of the pandemic, our results also highlight continued disparities in home versus institutional delivery across a range of socioeconomic indicators, specifically residence, wealth, education, and parity. After adjusting for other variables, residence was most strongly related to delivery location, with urban women being significantly more likely to deliver in a facility, and particularly a hospital, relative to rural women. Disparities in MNH care by residence are well-established [22, 23], but the strength of this association highlights the significant degree to which they persist in Ethiopia. While gains in institutional delivery are being made in both urban and rural areas, more rapid gains in urban centers may contribute to increased inequality within the country. These disparities are also reflected in the significantly increased odds of delivering in a facility among wealthier women; we found that even when stratified by residence, living in a less-poor household was significantly related to delivering in a facility. While there overall higher rates of facility delivery in urban areas, disparities between the poor and less poor in urban areas were particularly profound. Despite concerted efforts by the FMOH to reduce inequities in care [24], recent studies have found that inequity in MNH coverage by wealth may be increasing in Ethiopia [25, 26].

Women who were nulliparous prior to the index pregnancy were significantly more likely to deliver in a lower-level facility or a hospital than at home compared to multiparous women, even after adjusting for age. These results align with previous research in Ethiopia that found primiparous women were significantly more likely to use a skilled birth attendant and receive postnatal care services than higher parity women [27]. These differences may be due to a number of reasons, including greater confidence in the ability to successfully deliver among multiparous women, greater perceived risk of delivery among nulliparous women, or poor experience with the health system, but there is little research that explores this question explicitly. Previous research in Ethiopia found that motivation to deliver at home and with traditional birth attendants is largely driven by the belief that delivering in a facility is not necessary or customary [28], but whether this differed by parity was not explored. There is substantial evidence that low quality and disrespectful care is prevalent and a concern among women in Ethiopia [28–30], but longitudinal studies that assess whether the experience of low-quality care in a previous birth later negatively impacts the likelihood of delivering in a facility are limited. Understanding these motivations and how they vary by parity is critical in order to develop effective interventions.

PMA Ethiopia is unique in that COVID-19 occurred in the midst of ongoing data collection, thereby functioning as a "natural experiment", with "exposed" (those who delivered after COVID-19) and "unexposed" (those who delivered prior to COVID-19) cohorts. Our study is distinctly positioned to isolate the effect of COVID-19 on patterns of delivery location among a cohort of women in Ethiopia through a pre/post comparison. Additionally, while data collection was delayed by approximately two months due to COVID-19, PMA Ethiopia was able to complete interviews within an average of two months of delivery among women, limiting potential recall bias. Our study is not without limitations, however, primarily due to sample size considerations. As only 16% of our sample gave birth after the imposition of COVID-19 restrictions, we were restricted in our ability to explore disparities using nuanced categories for our independent variables. Additionally, while we included questions that explored how

COVID-19 impacted delivery location, we were limited in our ability to explore this fully as we were unable to conduct in-depth qualitative follow-up.

## Conclusion

We found that COVID-19 significantly affected delivery patterns among women living in urban areas, with shifts away from hospitals. This trend is a potential cause for concern, as lower-level facilities are generally less equipped to manage obstetric emergencies and provide high-quality obstetric care than hospitals. Concerns about COVID-19 transmission in health facilities and ensuring lower-level facilities are equipped to address obstetric emergencies will be critical to address as the pandemic continues. At the national level and in rural areas, however, there was no evidence of large-scale declines in rates of facility-based delivery. While COVID-19 will likely slow the considerable progress Ethiopia has made in increasing rates of institutional delivery and skilled birth attendance, modeled estimates that assume large declines in coverage of facility-based delivery and other key MNH interventions at the national level in Ethiopia are likely to overestimate the effect of COVID-19 on maternal health outcomes.

## Supporting information

**S1 Table. Reasons given for how COVID-19 affected delivery location.**
(PDF)

## Acknowledgments

We thank the PMA Ethiopia team for their tireless efforts to field the PMA survey, before and during the COVID-19 pandemic and the many respondents that contributed their time and responses to these surveys. We would also like to recognize the more than four million people who have lost their lives to COVID-19.

## Author Contributions

**Conceptualization:** Linnea A. Zimmerman, Selamawit Desta, Saifuddin Ahmed.

**Formal analysis:** Linnea A. Zimmerman, Saifuddin Ahmed.

**Funding acquisition:** Linnea A. Zimmerman, Selamawit Desta, Assefa Seme, Solomon Shiferaw, Saifuddin Ahmed.

**Investigation:** Linnea A. Zimmerman, Selamawit Desta, Assefa Seme, Solomon Shiferaw.

**Methodology:** Linnea A. Zimmerman, Celia Karp, Assefa Seme, Solomon Shiferaw.

**Project administration:** Selamawit Desta, Mahari Yihdego, Assefa Seme, Solomon Shiferaw.

**Supervision:** Linnea A. Zimmerman, Mahari Yihdego, Assefa Seme, Solomon Shiferaw, Saifuddin Ahmed.

**Visualization:** Linnea A. Zimmerman.

**Writing – original draft:** Linnea A. Zimmerman, Selamawit Desta, Celia Karp, Mahari Yihdego.

**Writing – review & editing:** Linnea A. Zimmerman, Selamawit Desta, Celia Karp, Mahari Yihdego, Assefa Seme, Solomon Shiferaw, Saifuddin Ahmed.

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
