## [Decision Letter · Decision Letter 0]

9 Aug 2021

 PGPH-D-21-00257 Effect of the COVID-19 pandemic on health facility delivery in Ethiopia; results from PMA Ethiopia’s longitudinal panel PLOS Global Public Health

Dear Dr. Zimmerman,

Thank you for submitting your manuscript to PLOS Global Public Health. After careful consideration, we feel that it has merit but does not fully meet PLOS Global Public Health’s publication criteria as it currently stands. Therefore, we invite you to submit a revised version of the manuscript that addresses the points raised during the review process.

We look forward to receiving your revised manuscript.

Kind regards,

Nicola Hawley

Academic Editor

Journal Requirements:

Additional Editor Comments (if provided):

Reviewers' comments:

Reviewer's Responses to Questions

**Comments to the Author**

1. Does this manuscript meet PLOS Global Public Health’s publication criteria? Is the manuscript technically sound, and do the data support the conclusions? The manuscript must describe methodologically and ethically rigorous research with conclusions that are appropriately drawn based on the data presented.

Reviewer #1: Yes

Reviewer #2: Yes

2. Has the statistical analysis been performed appropriately and rigorously?

Reviewer #1: Yes

Reviewer #2: Yes

3. Have the authors made all data underlying the findings in their manuscript fully available (please refer to the Data Availability Statement at the start of the manuscript PDF file)?

Reviewer #1: Yes

Reviewer #2: Yes

4. Is the manuscript presented in an intelligible fashion and written in standard English?

Reviewer #1: Yes

Reviewer #2: Yes

5. Review Comments to the Author

Reviewer #1: Dear Authors,

thank you very much for the interesting and important work.

Very few comments:

In the introduction you could as well relate to the MMR in Sierra Leone during the Ebola Crisis - to emphasise the importance of "side Effects" of an empidemic.

On page 14 in the 4th last line you used the verb developing instead of delivering.

Reviewer #2: Review Comments:

Abstract, settings sub-section

Covering 91% of the population. However, on page 10, dependent variable sub-section, the footnote shows that the PMA Ethiopia conducted in 6 regions with a total coverage of 90% of the population of the country. So, which figure is indicated the coverage rate of your population frame?

Independent variables sub-section, page 10/11

The variable of primary interest, which has a binary nature, of woman delivered prior to or during COVID-19 pandemic. Is it your variable of interest (dependent variable) or independent?

Since, the independent variables are listed as covariates (age, education, wealth …). To mean that independent variables are covariates.

Table 1, p-value=0.00, it better to use, p-value<=0.01

Table 1, rural row vs urban column, the figure shows 100, recheck it again, maybe it is 0.0.

6. PLOS authors have the option to publish the peer review history of their article (what does this mean?). If published, this will include your full peer review and any attached files.

**Do you want your identity to be public for this peer review?** For information about this choice, including consent withdrawal, please see our Privacy Policy.

Reviewer #1: No

Reviewer #2: No

---

## [Editor Report · Decision Letter 1]

15 Sep 2021

Effect of the COVID-19 pandemic on health facility delivery in Ethiopia; results from PMA Ethiopia’s longitudinal panel

PGPH-D-21-00257R1

Dear Dr. Zimmerman,

We're pleased to inform you that your manuscript has been judged scientifically suitable for publication and will be formally accepted for publication once it meets all outstanding technical requirements.

Within one week, you'll receive an e-mail detailing the required amendments. When these have been addressed, you'll receive a formal acceptance letter and your manuscript will be scheduled for publication.

An invoice for payment will follow shortly after the formal acceptance. To ensure an efficient process, please log into Editorial Manager at https://www.editorialmanager.com/pgph/ click the 'Update My Information' link at the top of the page, and double check that your user information is up-to-date. If you have any billing related questions, please contact our Author Billing department directly at authorbilling@plos.org.

Kind regards,

Nicola Hawley

Academic Editor